# Influence of Glass Silica Waste Nano Powder on the Mechanical and Microstructure Properties of Alkali-Activated Mortars

**DOI:** 10.3390/nano10020324

**Published:** 2020-02-14

**Authors:** Mostafa Samadi, Kwok Wei Shah, Ghasan Fahim Huseien, Nor Hasanah Abdul Shukor Lim

**Affiliations:** 1Faculty of Engineering, University Teknologi Malaysia, Johor Bahru 81310, Malaysia; kouchaksaraei@yahoo.com (M.S.); norhasanah@utm.my (N.H.A.S.L.); 2Department of Building, School of Design and Environment, National University of Singapore, Singapore 117566, Singapore; bdgv185@nus.edu.sg

**Keywords:** nanotechnology, waste glass, alkali-activated mortar, microstructure properties

## Abstract

The recycling of millions of tons of glass bottle waste produced each year is far from optimal. In the present work, ground blast furnace slag (GBFS) was substituted in fly ash-based alkali-activated mortars (AAMs) for the purpose of preparing glass bottle waste nano-powder (BGWNP). The AAMs mixed with BGWNP were subsequently subjected to assessment in terms of their energy consumption, economic viability, and mechanical and chemical qualities. Besides affording AAMs better mechanical qualities and making them more durable, waste recycling was also observed to diminish the emissions of carbon dioxide. A more than 6% decrease in carbon dioxide emissions, an over 16% increase in compressive strength, better durability and lower water absorption were demonstrated by AAM consisting of 5% BGWNP as a GBFS substitute. By contrast, lower strength was exhibited by AAM comprising 10% BGWNP. The conclusion reached was that the AAMs produced with BGWNP attenuated the effects of global warming and thus were environmentally advantageous. This could mean that glass waste, inadequate for reuse in glass manufacturing, could be given a second life rather than being disposed of in landfills, which is significant as concrete remains the most commonplace synthetic material throughout the world.

## 1. Introduction

The demand for ordinary Portland cement (OPC) is growing due to the ongoing proliferation of urban developments at the global level. Thus, within the concrete industry, OPC is currently the main binding agent [1,2,3]. However, the production of OPC is associated with high emissions of carbon dioxide, with one ton of carbon dioxide emitted for each ton of OPC that is produced [4,5,6]. Given that carbon dioxide contributes massively to the formation of greenhouse gases that promote global warming, OPC is not a sustainable material [7]. Better living standards have been achieved globally, thanks to the advances in industry and urbanism. However, these advances have been accompanied by issues of how to best manage industrial and domestic waste [7,8]. This has prompted worldwide initiatives to make waste management more effective, particularly in terms of waste recycling and reuse, as well as to search for viable alternatives to produce sustainable materials (e.g., ‘green’ concrete, which are materials produced from recycling of landfill waste) [9,10,11,12].

Cement-free materials that have recently become available include alkali-activated mortars (AAMs) and concretes. The preparation of these materials is undertaken based on alkali activation and starting source materials with high levels of silicon, aluminium and calcium [13,14,15], such as meta-kaolin (MK), fly ash (FA), palm oil fuel ash (POFA), GBFS and ceramic waste (WCP) [16,17,18]. Previous studies on AAMs revealed that these materials demonstrated a range of exceptional qualities, including short setting timespan, curing at room temperature [19], high early strength [20], low susceptibility to high temperatures, adequate durability in adverse conditions [21], reduced emissions of carbon dioxide, and low use of energy [22,23]. AAMs are usually produced with alkaline activator solutions like sodium hydroxide and sodium silicate, with conditions for ideal performance being high sodium hydroxide levels of 10–16 M and a high sodium silicate-sodium hydroxide ratio of 2.5 [24,25]. However, sodium silicate is not environmentally friendly, while sodium hydroxide does not only have an adverse environmental impact, but also poses a hazard to workers and is expensive [20,26,27]. Furthermore, the use of AAMs as novel materials for construction is hampered primarily by the high molarity displayed by sodium hydroxide and enriched sodium silicate in the alkaline solution. Environmentally speaking, this is worrisome as the synthesis of mineral-based AAM materials requires high levels of sodium silicate [28]. AAMs therefore have only a few uses within the construction sector, owing to such shortcomings related to alkaline solution. However, AAMs could be made stronger by incorporating calcium oxide from GBFS and other waste materials, even when the sodium hydroxide of low molarity (4 M) is used [29,30]. Moreover, the compatibility of C-(A)-S-H and N-A-S-H gels has notable implications for AAMs and alkaline solution-activated alumina-silicate systems, facilitating the synthesis of both products [31].

The preparation of AAMs as binding agents can be accomplished with various source materials. One especially promising material is FA as an industrial waste [32,33,34]. Rich in amorphous alumina and silica, FA represents a secondary product of coal burning in thermal power plants for electricity generation [5]. Its chemical constitution makes FA an appropriate source material for alkali-activated binder production. A number of studies have been dedicated to the investigation of the qualities of FA-based AAMs. AAMs are a promising cementitious material because they are highly durable and their suitability for different construction applications is supported by their confirmed engineering characteristics [21,35,36]. However, at high curing temperatures between 40 and 85 °C, FA-based AAMs present some shortcomings, such as long a duration of setting and poor compressive strength. These shortcomings could be improved by using the waste material known as ground blast furnace slag (GBFS) [27]. Calcium silicate hydrate (C-S-H) is the main reaction product of AAMs for GBFS, while amorphous hydrated alkali alumina-silicate is the main reaction product of AAMs for FA [20]. Despite its considerable strength, alkali-activated GBFS has restricted applications due to setting too fast, being inadequately workable and having high drying shrinkage values. FA-based AAMs could be made stronger, more workable and their setting time improved, while solution demand can be decreased by enriching them with GBFS [21]. However, GBFS makes mortar less durable in the context of exposure to sulphuric acid and sulphate attacks, primarily due to the fact that it has high levels of calcium oxide. Moreover, AAM mixtures can become more expensive and their energy consumption and carbon dioxide emissions can be heightened if the concentration of GBFS in the alkali-activated matrix is enhanced.

Concrete could be produced in a more sustainable manner by making use of glass waste products. At a global level, the amount of glass bottles disposed of every year figures in the millions of tons [37,38]. While a portion of the glass waste is reused, color differences, glass flaws and processing expenditure prevent the recycling of this waste in its entirety. Investigating the possibility of producing concrete, based on the use of glass, is not recent [38,39]. It has become common knowledge that ample proportions of seemingly non-crystalline aluminium and silicon could be present in bottle-derived pulverised glass waste. Thus, glass waste could be a potential pozzolanic or cement-like substance, so employing it in the form of aggregates to produce cement could be a viable option to actual cement. On the downside, the properties of the end-products could be altered by the use of glass waste [40].

The incorporation of nanomaterials, affording improved qualities, regardless of whether they are fresh or cured, is prioritized by current methods of concrete production. Nanoparticles (e.g., titanium dioxide, silicon dioxide, aluminium oxide, iron oxide, carbon nanotubes/fibres, and nano silica) are capable of filling pores and have a positive pozzolanic reaction, which is why they are widely employed to make concrete and cement perform better [41,42,43,44,45]. At the same time, fast infrastructural development at a global level has led to a growing demand for hybrid materials resembling cement. Such materials demonstrate advantages such as high strength, durability, sustainability and minimum environmental impact. Even high-grade concrete can perform significantly better when nano-silica is added. In fact, the addition of a proportion of 6% nano-silica is enough to attain marked increases in performance [43,46]. The incorporation of nano-silica affects mainly the process of hydration, which is sped up, fostering a greater production of calcium silicate hydrates as nano-silica reacts with the calcium hydroxide in the concrete. The outcome is improved mechanical properties.

Microstructures of greater compaction are formed in concrete enriched with nano-silica as the content of calcium hydroxide crystals is lower [47,48]. It has been reported that concrete supplementation with nano-silica increased the rate of the pozzolanic reaction by about 3% [49]. Furthermore, concrete enriched with nano-silica has been observed to be substantially denser, more durable, have higher tensile, compressive and bending strength, as well as to show less susceptibility to abrasion. Nano-silica also makes concrete less permeable and diminishes capillary absorption. In addition, improved splitting tensile strength and hydration speed have been observed in concrete enriched with nano-silica and ground GBFS [49,50].

It is clear that the enrichment of mortars with waste nanomaterials can have enormous potential. Therefore, the purpose of the present work was to substitute a small quantity of GBFS with BGWNP to produce AAMs, demonstrating effective performance, durability and sustainability. Glass bottle waste was used for preparation of the nano powder, which was then subjected to chemical and physical analysis. The parameters of mortar workability, setting duration, strength and microstructure framed the evaluation of the effect of BGWNP substitution of GBFS in AAM matrix.

## 2. Materials and Methods

The source materials employed for preparation of ternary AAM blend included FA, GBFS and BGWNP. Procured from the Tanjung Bin power station in Johor, Malaysia, FA was the primary aluminium-silicate source and was used untreated. The pure cement-free binding agent employed was GBFS, obtained from Ipoh (Malaysia) in ground form to attain the necessary dimensions of particles. Calcium and silicate were mainly sourced from this GBFS. Discarded glass bottles were obtained from Skudai (Malaysia). Their processing involved initial cleaning with normal tap water to eliminate contaminants prior to introduction in the crusher machine for crushing. Large glass particles were subsequently filtered by passing the crushed glass through a 600 µm sieve. To obtain 25 µm particles, the sieved glass was ground for three hours in a 25 kg volume Los Angeles Abrasion Machine with 16 stainless balls (diameter: 40 mm). The powder produced in this way was subjected to one-hour oven heating at 110 °C (±5), followed by seven-hour grinding in a ball mill machine to ensure that the nanoparticles were spread effectively. The process through which nano-powder was obtained from glass bottles is illustrated in Figure 1.

Aluminium silicate was activated through the mixture of sodium hydroxide with sodium silicate, thus obtaining the alkaline activator solution. Quality Reagent Chemical (QREC, Malaysia) supplied the sodium hydroxide of high purity (98%) and sodium silicate. Preparation of the alkaline activator solution was undertaken with low molarity sodium hydroxide (2 M) and a low ratio of sodium silicate to sodium hydroxide of 0.75, which remained constant for every solution concentration. These values were chosen to minimize the impact of sodium silicate and sodium hydroxide on the environment. Preparation of the sodium hydroxide solution involved dissolution of pellets in regular water, followed by cooling at ambient temperature for 24 h. Silicon dioxide, sodium oxide and water in proportions of 29.5 wt.%, 14.70 wt.% and 55.80 wt.%, respectively, constituted the solution of sodium silicate. The final alkaline solution was obtained by mixing the sodium hydroxide solution and the sodium silicate solution. Earlier research suggested that the ideal values were 20.75 wt.% for sodium oxide, 21.07 wt.% for silicon dioxide, and 58.2 wt.% for water for a sodium hydroxide solution of 14 M, with a ratio of sodium silicate to sodium hydroxide of 2.5 [26,51]. However, in the present work, different values were considered, namely, 10.53 wt.% for sodium oxide, 12.64 wt.% for silicon dioxide and 76.8 wt.% for water. The reason for choosing those values was that it was believed that this would yield an alkaline solution with a lower environmental impact and fewer emissions of carbon dioxide and more effective in terms of cost and energy consumption. For silicon oxide:sodium oxide, the solution had a final modulus (Ms) of 1.2. Furthermore, every mortar sample was produced by using natural siliceous river sand as a fine aggregate. The procedure began by washing the sand with water for removal of silts and impurities, following ASTM C117. The next step was the oven drying of the sand for 24 h at 60 °C to regulate the levels of moisture. Last but not least, grading was applied to the sand to ensure it complied with ASTM C33–33M.

The preparation of various mix proportions of GBFS: BGWNP was undertaken in different ratios, in line with ASTM C109–109M. For the control sample, preparation of the blend was undertaken through the mixture of FA and GBFS in a ratio of 70:30. For every mixture, the content of FA was kept at a constant 70 wt.%, while the content of GBFS was substituted with BGWNP by 5 wt.%, 10 wt.%, 15 wt.% and 20 wt.%. As shown in Table 1, for every substitution level, the ratio of binding agent to fine aggregate (B:A) was 1.0, the ratio of alkaline solution to binding agent (S:B) was 0.40, the NH molarity was 2, the ratio of NS to NH was 0.75, and the Ms was 1.2 by mass weight. ASTM C579 was followed when testing was performed. Compressive, flexural and splitting tensile strength were respectively evaluated based on a 50 mm cube mould, a 40 × 40 × 160 mm prism, and a 150 × 75-mm cylinder. To make demoulding more straightforward, motor oil was used for coating the inner parts of the moulds before casting. After mixing of the NS solution and NH solution, the mixture was allowed to cool to room temperature prior to use, which helped to prevent problems with the heat generated by the mixing.

BGWNP, GBFS and FA were mixed three minutes to prepare AAMs. This mixing process generated a homogeneous dry substance, which was subjected to additional mixing with fine aggregate for four minutes. The alkaline solution was subsequently added to activate the developed product, followed by medium-speed blending in a machine for five minutes. Last but not least, the produced mortar was introduced into the molds via a two-layer pouring technique. Air pockets were removed from the mixture by applying 15 s vibration to every layer. Completion of the process of casting was followed by 24 h curing of the AAMs at 24 ±1.5 °C and 75% relative humidity prior to demolding.

The chemical structure and the physical and mineral characteristics of raw materials affected the qualities of AAMs. An XRF test was conducted in this work to gain further insight into how BGWNP, GBFS and FA were chemically constituted. Chemical compounds, including silicon dioxide, aluminium oxide, calcium oxide, magnesium oxide, potassium oxide and sodium oxide, were assessed in terms of their percentage weight. Furthermore, PSA and TEM tests were carried out to investigate physical qualities, including the diameter size of the particles, while a visual test permitted examination of the color of the raw materials. The production of C,N-(A)-S-H gels and raw material reactivity depended significantly on the crystalline or amorphous phases of the raw materials, which were revealed by the XRD patterns.

A number of tests were conducted to investigate how the qualities of fresh and hardened AAMs were affected by nano-particles from glass bottle waste. For instance, the impact of GBFS substitution with BGWNP on AAM flowability was assessed via the flow test, and assessment of the setting time was undertaken as well. These two tests afforded valuable insight into the impact of BGWNP on the qualities of fresh AAMs. The sustainability of concrete products was closely reflected by the influence of supplementary materials on strength performance. Therefore, the compressive, flexural and splitting tensile strength of AAMs containing BGWNP instead of GBFS were evaluated via a number of tests. Additionally, to shed more light on the strength performance, microstructure tests (e.g., TGA, FTIR, SEM, XRD) were also carried out.

According to ASTM C230, assessment of the as-prepared AAM workability was conducted to understand the impact of BGWNP substitution of GBFS. The performance of fresh AAMs was investigated as well, and the flow diameter was measured. The AAM setting time was calculated through the Vicat needle test, which involved inserting a metal needle measuring 1.13 mm into the fresh samples at 24 °C room temperature and recording the level at which the needle penetrated. It was considered that the setting time was the time that passed between the mold filling termination and sample penetration at a depth of 0.5 mm. The mean value of 3 specimens from each AAM represented the setting time. Moreover, the flexural, splitting tensile and compressive strength of the AAMs were investigated in line with ASTM C78, ASTM C496/C496M-11 and ASTM C109/109M, respectively. The samples were cured for 28 days, and at the end of this, the final results were established as the mean of the results from 3 specimens of each AAM. The results derived from the strength tests were compared to the control sample.

Extraction and pulverisation of the middle portion of the AAMs were conducted after 28 days of curing. TGA, FTIR, SEM and XRD methods were employed to analyze the microstructure of the generated powder. The disorganized phase of the AAMs was validated by the XRD test conducted with the MDI Jade software (version 6.5). The results of the scanned samples were examined across the 2Ɵ range of 5–90°, 0.02 steps and 0.5 s/step scan speed. After they were placed on sample holders, the AAMs were dried through exposure to infrared radiation for five minutes and they were subsequently covered with gold by a Blazer sputter coater. Monitoring of the ensuring patterns was conducted at 20 kV and 1000× magnification. ASTM C140–07 tests were carried out on the AAMs after they were immersed in water at 24 °C for 24 h. Water was eliminated from the AAMs surface to determine their saturated weight (Ws). After drying in the oven at 105 °C for 24 h, the AAMs were measured for their dry mass (Wd) and water absorption (Wa) via the total immersion method was computed through Equation (1).
(1)$$\mathrm{Wa}\left(\%\right)=\frac{\mathrm{Ws}-\mathrm{Wd}}{\mathrm{Wd}}\;\times 100$$

## 3. Results and Discussion

### 3.1. Chemical and Physical Properties

The chemical constitution and physical qualities of BGWNP, GBFS and FA yielded by X-ray fluorescence spectroscopy are provided in Table 2. Aluminum and silica were established to be the main oxide compounds, occurring in BGWNP, GBFS and FA in proportions of 83%, 41.7% and 86%, respectively. By comparison to BGWNP and FA, GBFS had a considerably higher concentration of calcium oxide (51.8%). Specimen production depended to a great extent on the amounts of aluminum, silicate and calcium oxide, as these generated N, C-(A)-S-H gels within the process of geopolymerization. Meanwhile, in all three AAMs, the content of potassium oxide did not exceed 1%. GBFS had a higher content of sodium oxide (0.45%) than BGWNP (0.01%) and FA (0.08%). Potassium oxide and sodium oxide have a significant effect on the activation of geopolymerization and alkaline processes, as evidenced by earlier research. Moreover, in accordance with ASTM C618, the LOI values were low for BGWNP, GBFS and FA. In regards to the average size of particles, this was 10,000 nm for FA, 12,800 nm for GBFS and 80 nm for BGWNP (Table 2). Additionally, in FA and GBFS, no particle was larger than 45 µm, while in BGWNP, all particles were larger than 1 µm. With respect to color, FA was grey, GBFS was off-white and BGWNP was light-grey.

### 3.2. XRD Patterns of Raw Materials

The XRD patterns associated with BGWNP, GBFS and FA are shown in Figure 2. The occurrence of crystalline alumina and silica was considered to be the reason for the sharp peaks exhibited by FA in the 2θ range of 16–30°, while the presence of crystalline mullite and quartz phases was correlated with other diffraction peaks. The amorphous nature of GBFS was validated by the absence of peaks. Meanwhile, GBFS development depends to a great extent on the occurrence of reactive calcium and silica substances. Indeed, GBFS could be optimal for the production of AAMs, precisely because of its high content of reactive amorphous calcium and silica. Considerable quantities of reactive aluminum and silica were found to be present in BGWNP, since the XRD pattern of this material highlighted its amorphous characteristics. In addition, the crystalline quartz peaks displayed by BGWNP at 27° and 29° were weak. This observation is consistent with the findings of Ismail et al. [52].

### 3.3. TEM Image of Specimens

BGWNP contained ellipsoidal nanoparticles, as revealed by TEM imaging (Figure 3). High surface tension and the surface energy of nanoparticles of heightened refinement were believed to be the reason for the aggregation of smaller nanoparticles into larger ones. The catalytic activity of the nanoparticles was enhanced by the broad surface area generated by the small nanoparticles. The obtained nano-crystallite was, on average, 80 nm in size, which was consistent with the nano-crystallite in the compound based on nano-silica. Furthermore, BGWNP displayed fluctuating super-elasticity, with a composition comparable to that observed in other studies.

### 3.4. Workability

Figure 4 shows how the workability of ternary blended AAMs was impacted by BGWNP replacing GBFS. The rise in the level of BGWNP made AAMs less workable. The original flow diameter of 15.5 cm at zero nano-powder content decreased to 15.3 cm at 5% nano-powder content, 14.7 cm at 10%, 14.2 cm at 15%, and 13.5 cm at 20% nano-powder content. As GBFS was substituted in the AAM matrix, the concentration of nano-powder increased, leading to a broadening of the specific surface area of the ternary binding agent [50]. In line with earlier results [53,54], the conclusion reached was that workability was adversely affected by a high specific area, as this increased the water requirement. Separation and bleeding were minimized, while the mixture was made more cohesive by the introduction of nano-materials. It was reported by Chithra et al. [55] that the addition of nanosilica in concrete decreases workability and this is attributed to the fact that some portion of the mixing water was absorbed by nanosilica particles.

The content of BGWNP affected the initial and final setting times of the AAMs, as shown in Figure 5. More specifically, as the level of BGWNP in the AAM matrix increased, the two setting times increased. Substitution of GBFS with BGWNP, from 0% to 20%, increased the initial setting time from 34 to 51 min and the final setting time from 52 to 76 min. Thus, the increase in BGWNP content led to a greater discrepancy between the two setting times. It was implied from this that the rate of setting increased with the reduction in the level of GBFS in the matrix [27]. Other studies [14,56,57] made comparable observations, with workability being noted to be improved while setting times were increased by a reduction in the ratio of calcium oxide to silicon dioxide. Nevertheless, in this work, the AAMs became more workable at a slow rate, despite the decrease in GBFS level from 30% to 10%. Hence, it was concluded that the qualities of AAMs were markedly influenced by BGWNP. There is evidence that the hydration rate can be sped up and the setting time diminished by silica and other nano-materials [55,58].

### 3.5. Compressive Strength

The compressive strength of AAMs cured for 28 days, in relation to the level of GBFS substitution with BGWNP, is indicated in Figure 6. There was a steady increase in compressive strength, from 56.2 to 65.5 MPa, as the BGWNP level was increased from 0% to 5%. However, an increase in the BGWNP level from 5% to 20% caused a decrease in compressive strength to 42.1 MPa. Despite the confirmed advantages of nano-silica addition to cementitious materials, a consensus is yet to be reached regarding the ideal proportion of nano-silica substitution. In this work, the way in which nano-silica was produced and the way in which BGWNP was spread in cementitious materials were the primary reasons for the strength increase. The pozzolanic reaction was chiefly intended to improve strength and diminish the distribution of pore sizes. The increase in the level of nano-silica up to 10% improved mortar compressive strength, but it decreased it at concentrations higher than 10% [48,55]. The results indicated that the compressive strength enhanced by including the nano-silica in AAMs mixtures, which is in agreement with the findings of previous studies [46,59,60]. Other research [41,43] indicated that reduced water assimilation and improved structural density, thus affording greater compressive strength, could be achieved by adding 4.0–6.0% nano-silica by the weight of FA. Nano-silica reacts with lime during the cement hydration process and it generates a C–S–H gel that may improve the mechanical strength and durability of concrete. A good dispersion of nano-silica into cement-based materials can enhance the hydration process of cement paste, allowing for a denser microstructure [61].

### 3.6. Flexural, Splitting Tensile Strength and Modulus of Elasticity

The impact of GBFS substitution with BGWNP on AAM flexural strength (FS) is illustrated in Figure 7. The AAMs were assessed for alterations in strength properties after they were cured for 28 days. It was observed that FS was inversely correlated with BGWNP when the level of the latter was no higher than 10%, while at the 0–5% BGWNP level, the FS increased by 14%. The FS improvement could be explained in terms of the fact that the occurrence of the nano-powder afforded the microstructure better hydration qualities. By contrast, a BGWNP level exceeding 10% diminished FS. More specifically, the FS was 7.2–6.4 MPa at 5–10% BGWNP and subsequently declined to 6.4–5.7 MPa at 10–15% BGWNP and 5.7–5.4 MPa at 15–20% BGWNP. At 15% BGWNP, the FS was compared with the control sample and was observed to have declined by 9.5%. This was consistent with the findings of earlier studies [62,63], which reported that BGWNP levels exceeding 10% had a negative effect on strength. A possible reason for this could be a high water requirement, which was why the hydration process was affected.

The impact of GBFS substitution with BGWNP on the splitting tensile strength (STS) of AAMs cured for 28 days is illustrated in Figure 8. The STS was 3.6 MPa in control samples with a ratio of GBFS to BGWNP of 30:0. At 0–5% BGWNP, AAMs displayed STS of 3.6–4.4 MPa, while at 5–10% BGWNP, they displayed STS of 4.4–4.8 MPa. By contrast, at 15% BGWNP, STS declined by 2.9 MPa, while at 20% BGWNP, STS declined by 2.8 MPa.

The impact of GBFS substitution with BGWNP on the modulus of elasticity (MoE) of AAMs cured for 28 days is illustrated in Figure 9. At 0–5% BGWNP, the MoE increased, but subsequently declined from 15.4 to 14.4 GPa, in comparison to the BGWNP-free control sample, which had a MoE of 14.1 GPa. By contrast, at 15–20% BGWNP, the MoE of the AAMs diminished from 13.8 to 13.6 GPa. The lower calcium content of AAMs with no more than 10% BGWNP was considered to be the reason for the reduction of not only MoE, but also of FS and STS [21].

The correlation among the FS, STS, MoE and compressive strength of the AAM samples is illustrated in Figure 10. Equations (2)–(4) reveal the correlation of the empirical data based on the exponential regression technique. The correlation had a good confidence level as the R2 values were in the range 0.87–0.99.
(2)$$\mathrm{FS}=3.2225e^{0.0122\mathrm{cs}}$$
(3)$$\mathrm{STS}=e^{0.0228\mathrm{cs}}$$
(4)$$\mathrm{MoE}=10.878e^{0.005\mathrm{cs}}$$

### 3.7. Water Absorption

The impact of BGWNP addition on the water absorption of AAMs cured for 28 days is shown in Figure 11. The increase in BGWNP up to 10% caused a reduction in water absorption. The ratio of GBFS substitution with BGWNP had a significant effect on AAM water absorption, which was 8.9 at 5% BGWNP, 9.6 at 10% BGWNP, 10.4 at 15% BGWNP and 10.6 at 20% BGWNP. In comparison to the control sample, water absorption was lower in the samples with 5% and 10% BGWNP, which exhibited a packed pore structure. However, at a BGWNP content exceeding 10%, water absorption was increased to 10.4 at 15% BGWNP and 10.6 at 20% BGWNP. There are several explanations for these results. The elevation in BGWNP concentration from 0% to 5% caused the formation of C-A-S-H gel of higher density, improving uniform strength with reduced water absorption. In comparison to control, the sample with 5% BGWNP was less porous (12.7%), suggesting that binder hydration was favorably influenced by BGWNP. The C-S-H phase, developing due to the BGWNP nucleation effect, improved binder hydration because it was not limited solely to the grain surface, while also filling a greater proportion of pores [63]. Therefore, at 5% and 10% BGWNP, AAMs initially became less porous. Nevertheless, as the level of BGWNP rose, so did the AAM air content. Thus, when the developed C-S-H could no longer make up for the pores produced by the air trapped in the fresh mortar, there was renewed increase in porosity. Hence, it was deduced that an ideal BGWNP level providing minimal AAM porosity existed.

### 3.8. XRD Pattern

The XRD patterns and crystalline structure of AAMs cured for 28 days are shown in Figure 12. The occurrence of AAM gel was signaled by an amorphous hallow between 20° and 35°. As the albite and gismondine peaks became more intense, the effect of BGWNP incorporation in the AAM matrix emerged in the XRD patterns between 24° and 34°. By contrast, BGWNP addition made the quartz peak at 36° less intense. These two observations implied the production of a greater amount of C, N-(A)-S-H gels, which improved hydration and geopolymerisation. 5% and 10% BGWNP gave the most intense albite and gismondine peaks, in comparison to the control sample. Meanwhile, 15% and 20% BGWNP weakened the strength by having an adverse effect on geopolymerization and diminishing the formation of C, N-(A)-S-H gels.

### 3.9. FESEM and EDX Analyses

Microstructural image analysis permitted the investigation of the effect of GBFS substitution with BGWNP in AAMs. FESEM imaging of AAMs is illustrated in Figure 13, while the energy dispersive X-ray (EDX) spectra of the AAMs are shown in Figure 14. BGWNP content of 5% and 10% enabled the samples to perform exceptionally, with limited porosity and unreacted particles (Figure 13a,b). The partial reaction of FA particles was revealed by the FESEM images, while a non-negligible amount of unreacted FA spheres was left in the AAM matrix. An incomplete crystalline structure, 150–300 nm in diameter, was observed alongside the amorphous reaction products. Furthermore, on the surface of and around the FA particles, there were needle-shaped crystals, some of which were nearly entirely coated in a dense amorphous gel layer. Jang and colleagues had previously observed such crystals as well [64]. On the whole, FESEM imaging confirmed that the AAMs consisted of a combination of unreacted fly ash particles, a few crystals and a reacted gel phase. Meanwhile, as can be seen in Figure 13c,d, 15% and 20% BGWNP content led to the emergence of a greater proportion of unreacted particles with a structure displaying poor density.

The addition of 5% BGWNP was revealed by the EDX spectra to result in a calcium oxide–silicon dioxide ratio of 1.15 (Figure 14); however, the content of calcium oxide in the matrix was diminished. By contrast, the addition of 15% BGWNP yielded a lower calcium oxide–silicon dioxide ratio of 0.64. Meanwhile, the addition of 5–15% BGWNP led to a rise in the silicon dioxide–aluminium oxide ratio from 1.74 to 2.55. The addition of 5% BGWNP manifested on EDX spectra in somewhat of a low silicon dioxide–aluminium oxide, which might suggest the replacement of higher aluminum ions in the C, N-A-S-H chain. Furthermore, the decrease in compressive strength, as the level of BGWNP was enhanced from 5% to 15%, could be due to the increase in the concentration of silicon dioxide alongside a decrease in the concentration of aluminum oxide and calcium oxide. Thus, by comparison to samples with 5% BGWNP, gel formation was more limited in samples with 5–15% BGWNP.

### 3.10. FTIR Analysis

The development of reaction products and the extent of geopolymerization in different AAM matrices were investigated based on FTIR analysis. As shown in Figure 15, the reaction sites of silicon-oxygen and aluminium-oxygen in AAM blends were identified by FTIR via chemical analysis, which enabled detection of functional groups on the basis of bonding vibrations. As minerals derived from the supplementation of base materials with alkaline activators dissolved, compressive strength developed in the matrix. Consequently, aluminum was released through hydroxylation, leading to the formation of Al-O-Al bond by the binding of -OH ions within the alkali. This was achieved through disruption of the weak bonds, releasing aluminum with a negative charge in IV fold coordination. Calcium demonstrated preferential reaction over sodium, thus accomplishing a balanced charge. GBFS had satisfactory potential for calcium solubility in the mixture because its content of calcium oxide was higher compared to FA. The compressive strength was influenced by the fact that the GBFS content determined the amount of soluble calcium. To harden the product, polycondensation could be performed to construct the unit oligomer of (-Si-O-Al) Ca in chains, sheets or the three-dimensional framework [51].

The present work sought to make AAMs stronger, more durable and more sustainable by substituting GBFS with BGWNP. Over the curing period of 28 days, the strength of AAMs increased from 56.2 to 65.5 MPa as the level of BGWNP was heightened from 0% to 5%. However, the strength decreased to 56.9 MPa at 10% BGWNP, 46.4 MPa at 15% BGWNP, and 42.1 MPa at 20% BGWNP (Figure 6). Meanwhile, an increase in C(N)-A-S-H gel was signaled by the reduction in the Si-O-Al band frequency. Consequently, at 5% and 10% BGWNP, the AAMs exhibited a highly uniform structure, as well as greater silicate reorganization, in comparison to samples without BGWNP. However, compressive strength decreased, while the band frequency increased to 990.4 cm^−1^ at 15% BGWNP and 995.4 cm^−1^ at 20% BGWNP. Furthermore, as BGWNP content was elevated from 0 to 5%, the Si-O-Si bending modes changed from 775.2 to 754.2 cm^−1^. Intensified production of C-S-H gel was reflected by the reduction in the Si-O-Si band frequency with BGWNP elevation. Moreover, as the molecular molar mass of the bound atoms increased, the vibration frequency declined. This decrease in vibrational frequency was due to the dislocation of Si atoms from the Si-O bonds caused by the release of soluble calcium by GBFS. However, both the ratio of silicon dioxide to aluminum oxide and the Si-O-Si (Al) vibrational frequency increased when BGWNP was added [65].

Evidence has been provided indicating that, in the context of calcium-based AAMs, the gehlenite phase is hydroxylated to calcium-ortho-sialate-disiloxo, while the akermanite phase is hydroxylated to calcium-disiloxonate hydrate (C-S-H), with condensation being the common outcome. Furthermore, samples containing 0%, 5% and 10% FA binding agent exhibited aluminate vibration band changes from 873.6 to 872.1 and 874.8 cm^−1^, respectively. This was indicative of structural reorganization as well. It was thus deduced that elevation of the concentration of BGWNP induced modifications in AAM structure, possibly due to the intensification of C-S-H and C(N)-A-S-H gel production and the increase in the quantity of nano-silica. The geopolymerization rate was retarded and the AAM mechanical strength was adversely impacted by such modifications.

### 3.11. TGA and DTG Analysis

The percentage of AAM weight loss was obtained via TGA and DTG analyses. The TGA and DTG curves of AAM with 5% BGWNP are shown in Figure 16a, while the TGA and DTG curves of AAM with 15% BGWNP are shown in Figure 16b. By comparison to AAM with 15% BGWNP, which had over 12.12% weight loss and stable behavior, AAM with 5% BGWNP had weight loss of just 10.47%. Moreover, AAM with 15% BGWNP contained less C-S-H gel than AAM with 5% BGWNP (7.82% vs. 9.63%). In contrast, AAM with 5% BGWNP had a lower percentage of calcium hydroxide than AAM with 15% BGWNP (4.53% vs. 5.73%). The great quantity of C-S-H gel and the reduced proportion of calcium hydroxide were considered to be the reasons why AAM with 5% BGWNP was more stable. Thus, it was concluded that BGWNP was useful for improving AAMs in terms of microstructure characteristics and strength performance. The capacity of hydration products to make a bulk paste matrix denser and to improve the mechanical qualities and durability of mortars with nano-powder content was confirmed by Singh and colleagues as well [66].
C-S-H gel (%) = Total LOI − LOI _CH_ − LOI _CC_(5)
where LOI(CH) is the dehydration of calcium hydroxide in the 400 to 550 °C range, and LOI(CC) is the carbon dioxide loss in the 600 to 750 °C range.
(6)$$\mathrm{CH}\;(\%)\;=\;\mathrm{WL}{\;}_{\mathrm{CH}}\;(\%)\;\times \;[\mathrm{MW}{\;}_{\mathrm{CH}}/\mathrm{MW}{\;}_{H}]$$
where WL(CH) is the weight loss ascribed to CH dehydration, MW(CH) is the molecular weights of CH (74 g.mol^−1^), and MW(H) is the molecular weight of water (18 g.mol^−1^).

## 4. Conclusions

The results of the present study confirm that substituting GBFS with GBWNP in AAM production is viable. Several conclusions have been derived from the empirical work:

First, the increase in the level of BGWNP decreased the AAMs working time as BGWNP incorporation has been shown to delay the setting.

Secondly, the addition of 5% BGWNP was established to be the most suitable to improve AAMs engineering properties. Still, for the replacement level of 10%, there was still engineering performance improvement in comparison to the reference mortar.

Thirdly, 5% and 10% BGWNP-containing AAMs became more durable and absorbed less water.

Last but not least, AAMs with a content of BGWNP of less than 10% exhibited an improved microstructure and C-S-H gel development, according to the results derived from XRD, FESEM, EDS, FTIR and TGA analyses.

## Figures and Tables

**Figure 1 nanomaterials-10-00324-f001:**
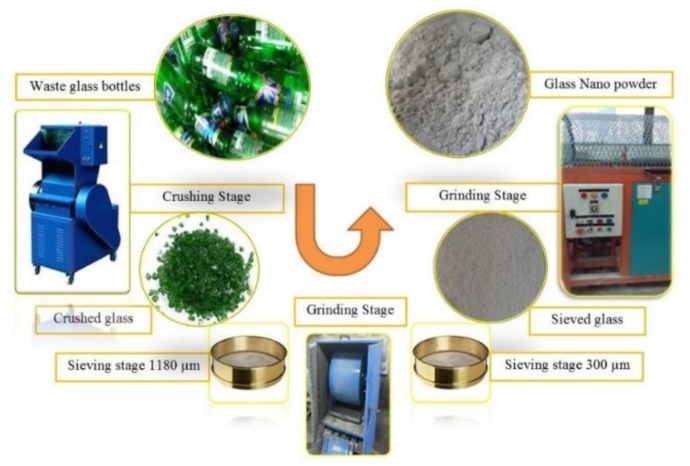
BGWNP preparation stages.

**Figure 2 nanomaterials-10-00324-f002:**
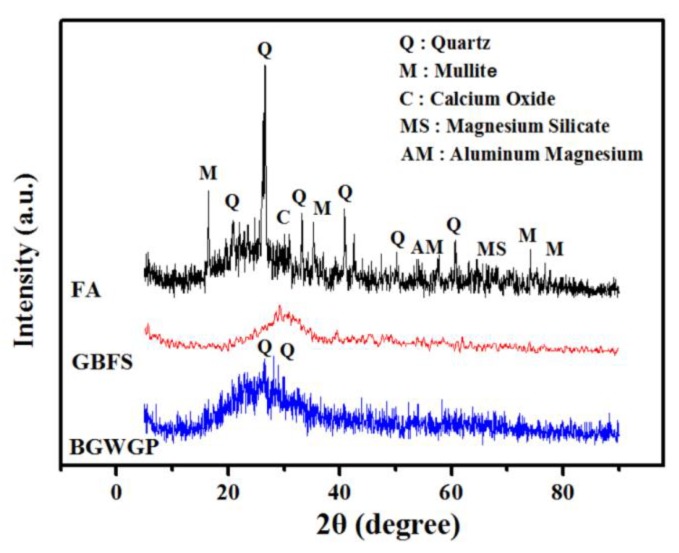
XRD patterns of FA, GBFS and BGWNP.

**Figure 3 nanomaterials-10-00324-f003:**
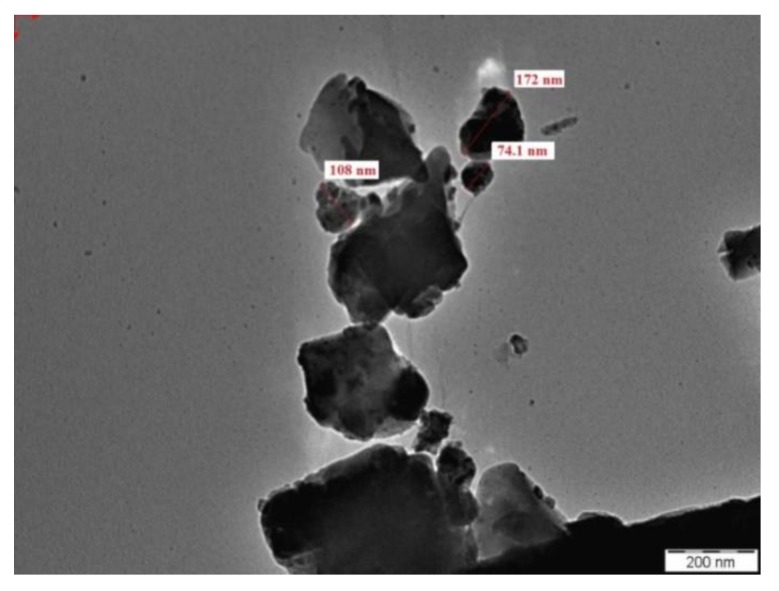
TEM image of BGWNP.

**Figure 4 nanomaterials-10-00324-f004:**
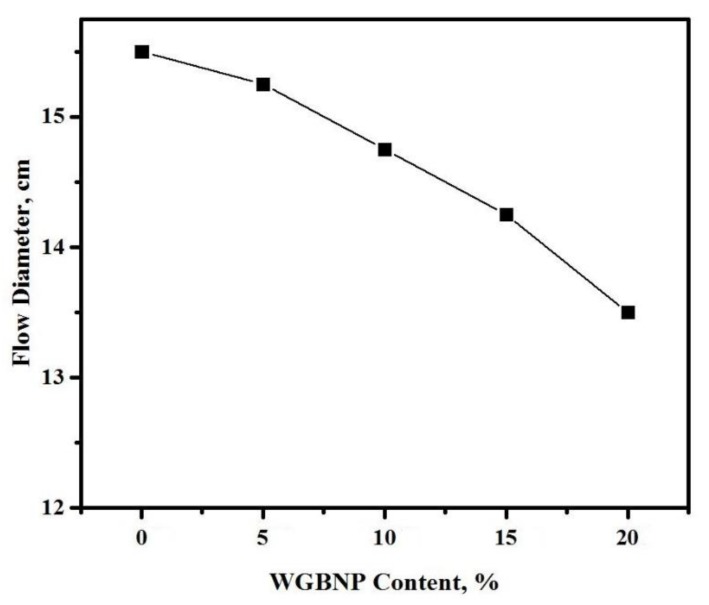
BGWNP content dependent flow diameter variation in prepared AAMs.

**Figure 5 nanomaterials-10-00324-f005:**
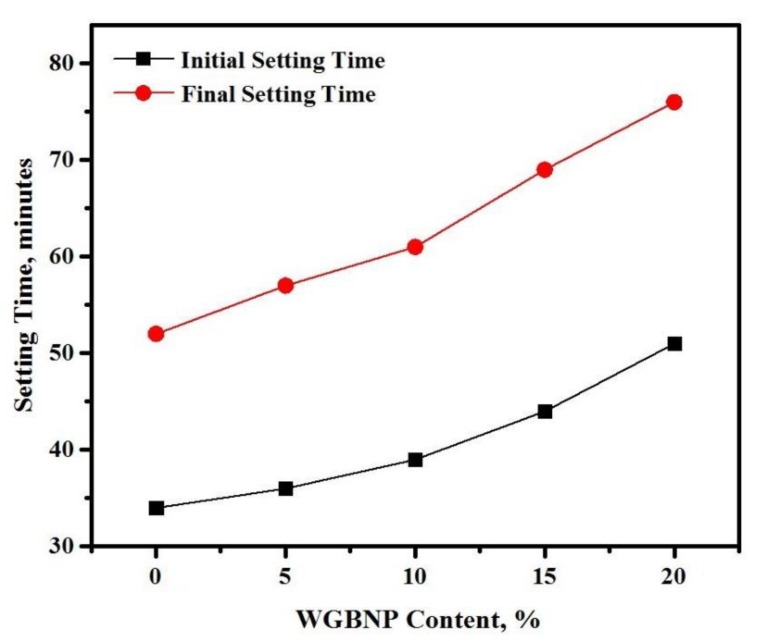
Setting times of prepared AAMs.

**Figure 6 nanomaterials-10-00324-f006:**
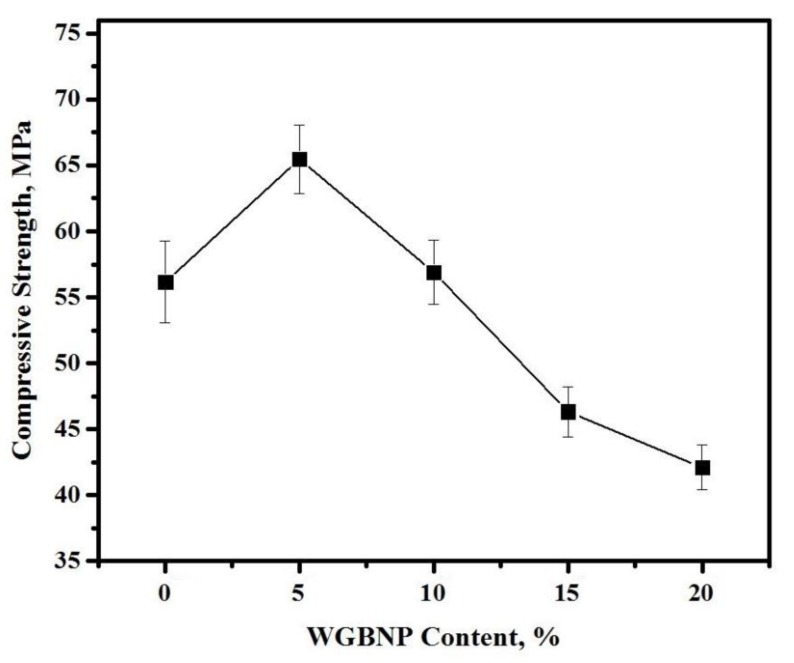
Effect of varying the BGWNP content on the compressive strength development of AAMs.

**Figure 7 nanomaterials-10-00324-f007:**
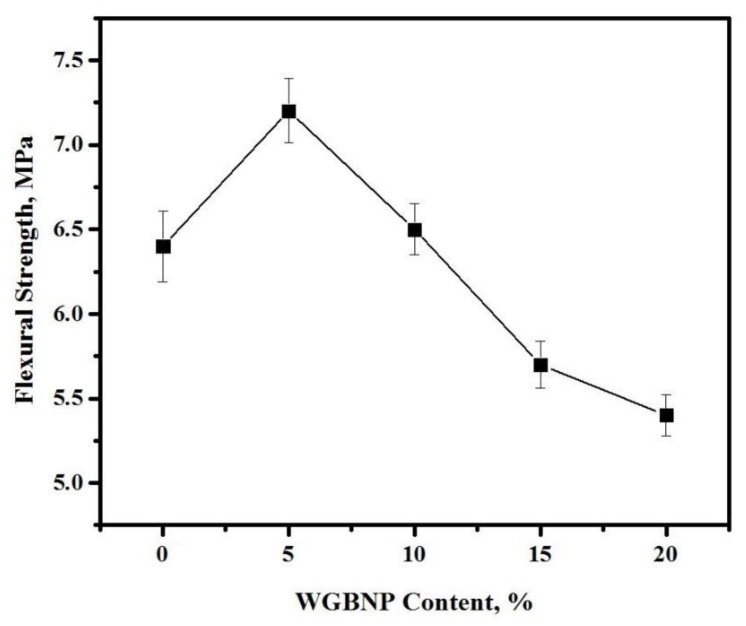
Influence of AAMs flexural strength by BGWNP content.

**Figure 8 nanomaterials-10-00324-f008:**
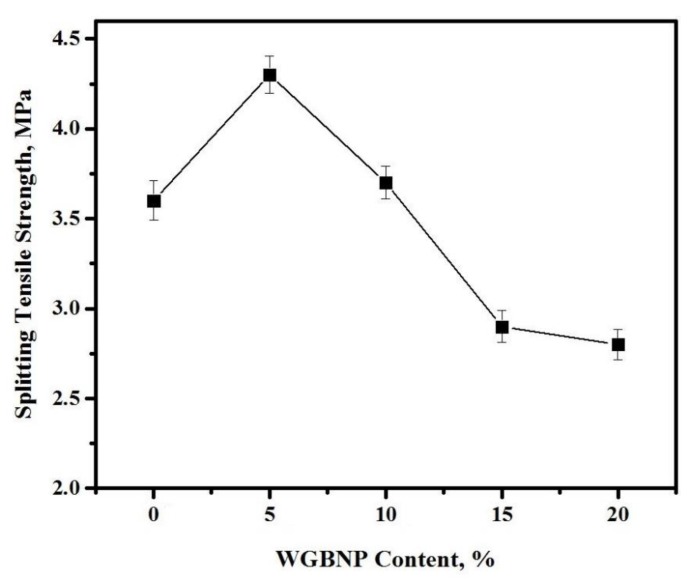
Influence of AAMs splitting tensile strength by BGWNP content.

**Figure 9 nanomaterials-10-00324-f009:**
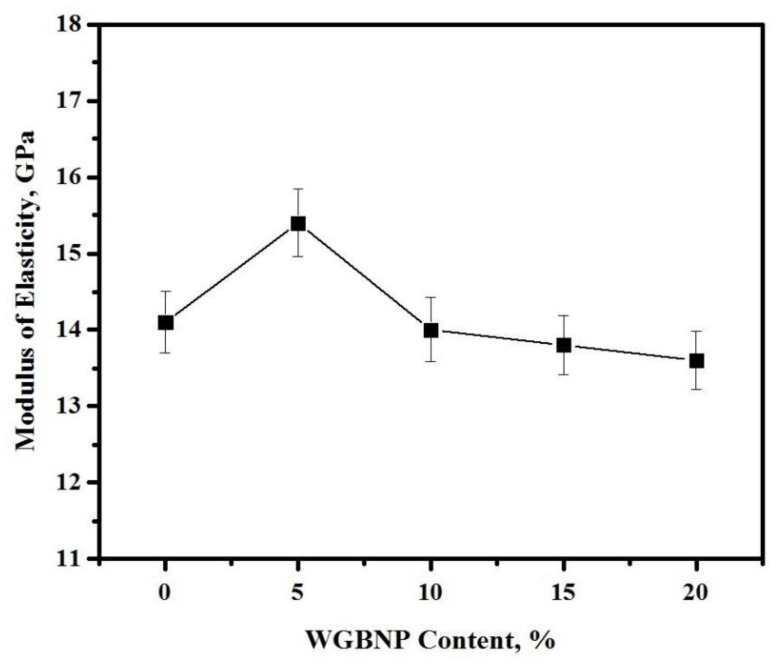
Influence of BGWNP content on MoE of AAMs.

**Figure 10 nanomaterials-10-00324-f010:**
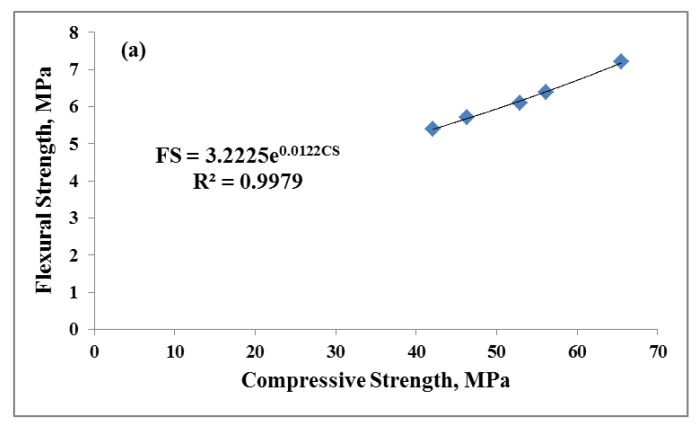
Relationship between CS (**a**) FS, (**b**) STS and (**c**) MoE of AAMs.

**Figure 11 nanomaterials-10-00324-f011:**
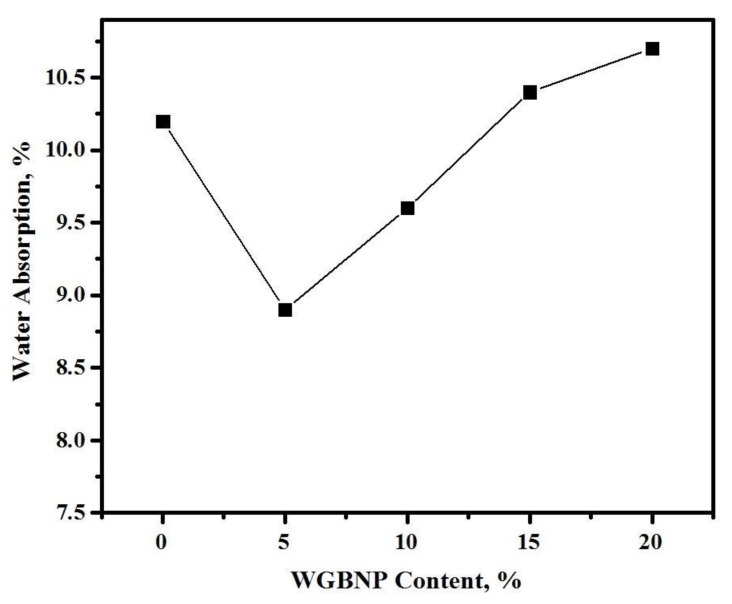
Water absorption of AAMs containing various ratios of BGWNP at 28 days.

**Figure 12 nanomaterials-10-00324-f012:**
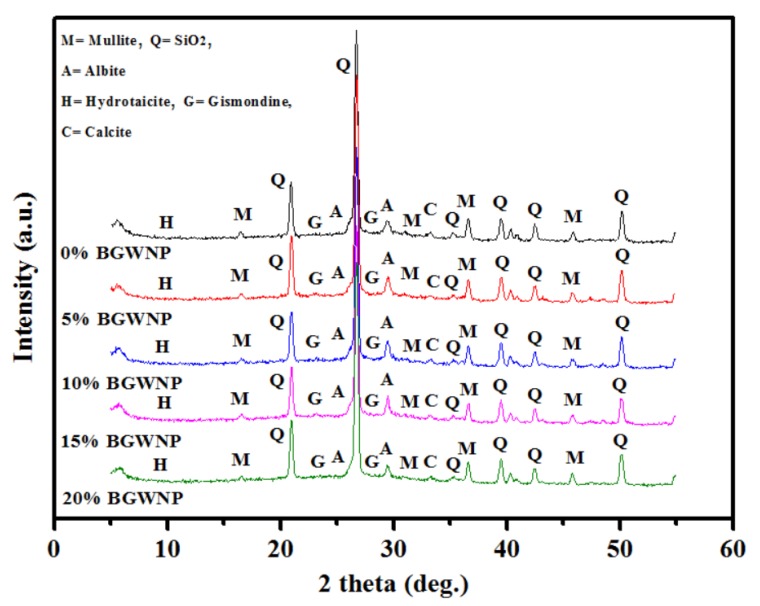
Effect of various BGWNP contents on structures of prepared AAMs.

**Figure 13 nanomaterials-10-00324-f013:**
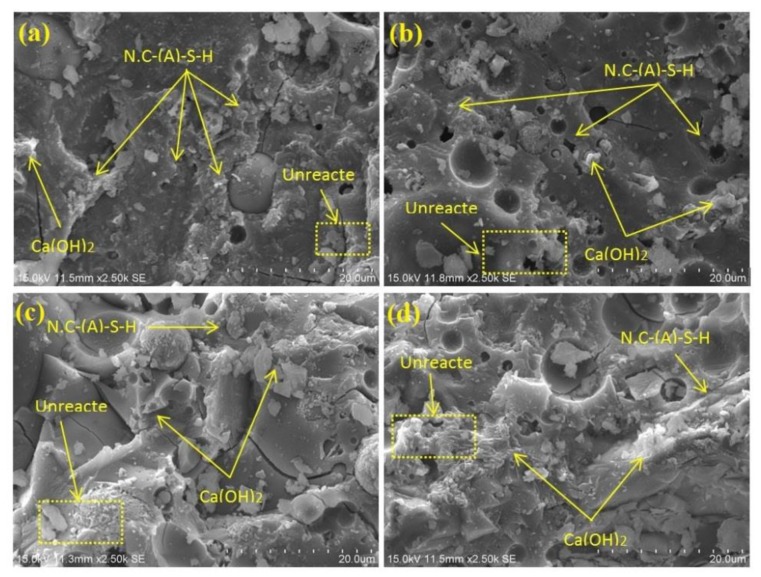
FESEM images of AAMs containing different proportions of BGWNP.

**Figure 14 nanomaterials-10-00324-f014:**
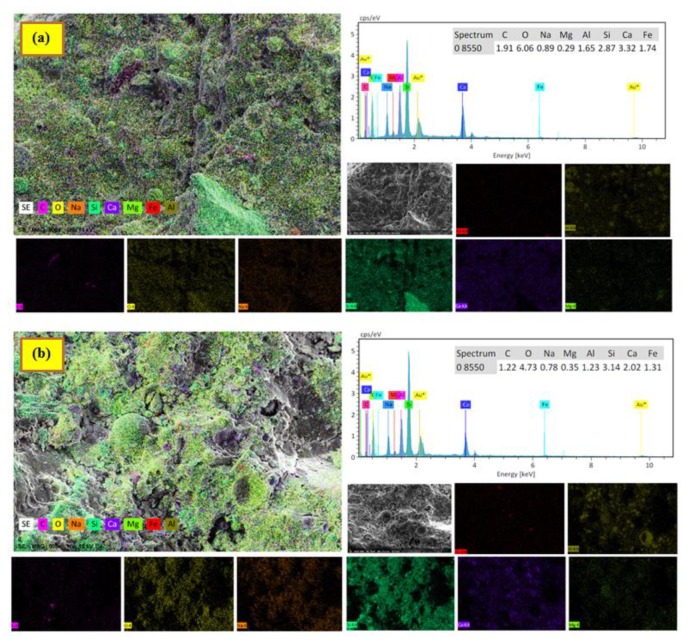
EDX spectra and maps of AAMs containing BGWNP at (**a**) 5% and (**b**) 15%.

**Figure 15 nanomaterials-10-00324-f015:**
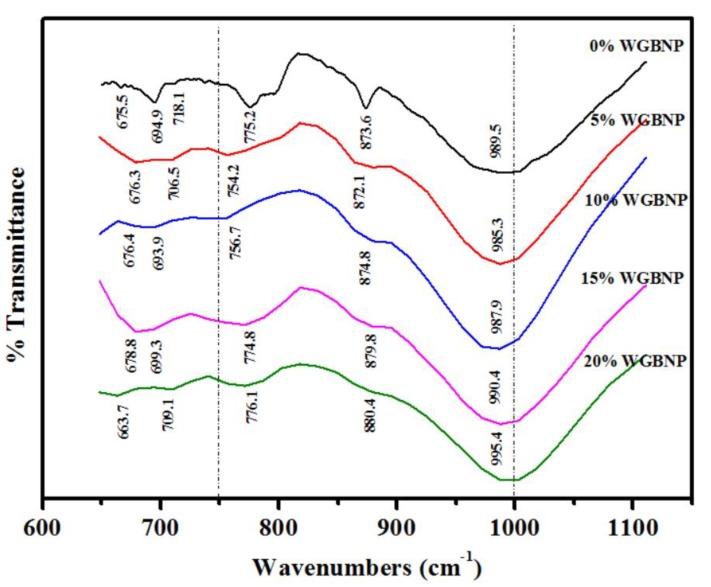
FTIR spectra of AAMs prepared with different amounts of BGWNP.

**Figure 16 nanomaterials-10-00324-f016:**
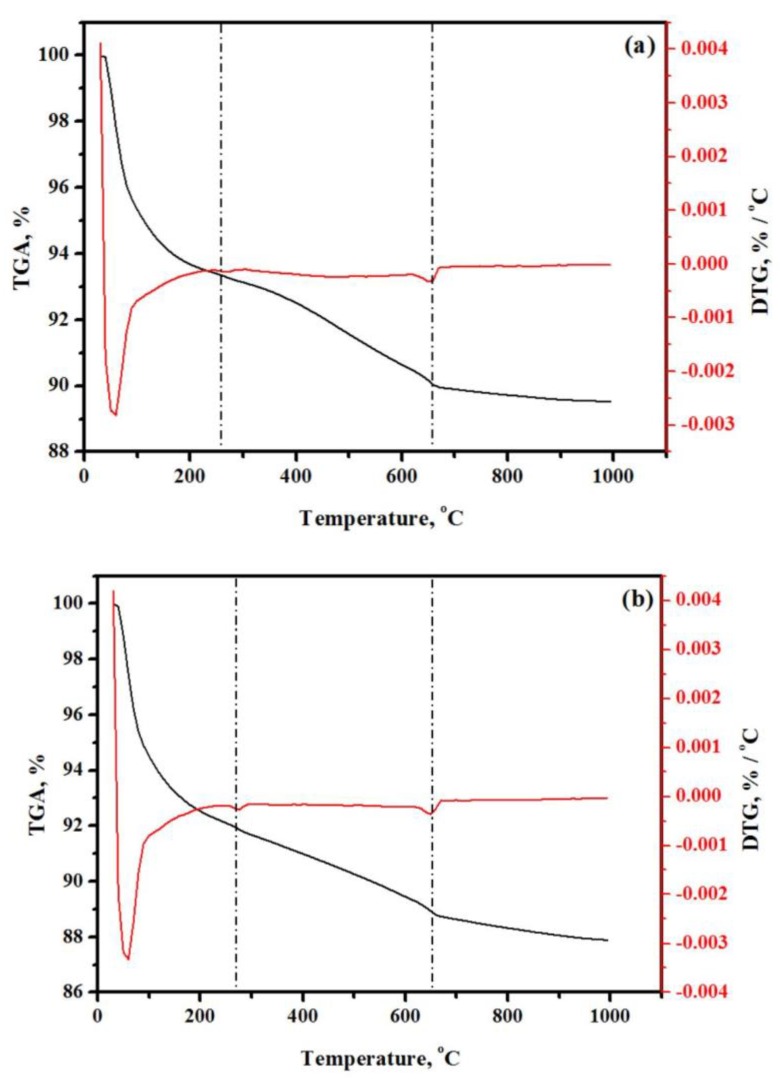
TGA and DTG curves of prepared AAMs with BGWNP at (**a**) 5% and (**b**) 15%.

**Table 1 nanomaterials-10-00324-t001:** Mix design of AAMs prepared with various ratios of BGWNP replacing GBFS.

		Mix Design Formulation of Alkali Activated Mortars				
Materials (mass, %)		AAMs_1_	AAMs_2_	AAMs_3_	AAMs_4_	AAMs_5_
Binder (B)	FA	70	70	70	70	70
	GBFS	30	25	20	15	10
	BGWNP	0	5	10	15	20
Binder:Fine aggregate (B:A)		1.0	1.0	1.0	1.0	1.0
Solution:Binder (S:B)		0.40	0.40	0.40	0.40	0.40
Na_2_SiO_3_:NaOH		0.75	0.75	0.75	0.75	0.75
Sodium hydroxide (NaOH)	Molarity, M	2.0	2.0	2.0	2.0	2.0
	H_2_O	92.6	92.6	92.6	92.6	92.6
	Na_2_O	7.4	7.4	7.4	7.4	7.4
Sodium silicate (Na_2_SiO_3_)	H_2_O	55.8	55.8	55.8	55.8	55.8
	Na_2_O	14.7	14.7	14.7	14.7	14.7
	SiO_2_	29.5	29.5	29.5	29.5	29.5
Total H_2_O in alkaline solution		76.8	76.8	76.8	76.8	76.8
Modulus of solution (Ms) SiO_2_:Na_2_O		1.2	1.2	1.2	1.2	1.2

**Table 2 nanomaterials-10-00324-t002:** Chemical and physical properties of FA, GBFS and BGWNP.

Material	FA	GBFS	BGWNP
SiO_2_	57.20	30.8	69.14
Al_2_O_3_	28.8	10.9	13.86
Fe_2_O_3_	3.67	0.64	0.24
CaO	5.16	51.8	3.16
MgO	1.48	4.57	0.68
K_2_O	0.94	0.36	0.01
Na_2_O	0.08	0.45	0.01
SO_3_	0.10	0.06	4.08
LOT	0.12	0.22	0.16
Others	2.45	0.20	8.66
Physical characteristics			
Average diameter (nm)	10,000	12,800	80
Colour	Gray	Off-white	Light-gray

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
