# Peer review of "Influence of Glass Silica Waste Nano Powder on the Mechanical and Microstructure Properties of Alkali-Activated Mortars"

_nanomaterials, 2020, doi:10.3390/nano10020324_

Round 1

Reviewer 1 Report

The article can be published after the following changes:

The discussion of the results is too weak and they are not compared with others results presented in papers using similar methods and recycled glass. The discussion part must be completed (expanded), and the results compared with those of other authors. The references should be prepared according to Journal Nanomaterials instructions for authors. Furthermore all authors in article should be mentioned. Chemical composition of FA and GBFS is not equal to 100% (table 1). Which was more in this materials? Chapter 3.7. (Statistical data analysis) needs improvement and clarification. Incorrect numbering of figures. Authors should be correct the axis descriptions on the graphs. There are two figure No. 12. Authors should improve numbering, also in the text. Authors should clearly explain abbreviations used in text. Authors should improve chapter 3 - Results and Discussion. In the subsections Authors should be describe in text the same results than on in figures.

Author Response

Thanks for suggestion and critical comments. response to reviewer as attached. 

Reviewer 2 Report

The subject of the research is interesting and is in a novel area as proved by having 55 references from the last decade, whit a great share for the last 3 years. The introduction duly justifies the research need and provides proper framing for the work.

The section concerning the experimental program is thoroughly detailed, being and added value for the manuscript. Moreover, the experimental program is quite comprehensive.

The major criticism on this work goes to section 3.7. The first one, because the numbering is repeated. Probably, a former reviewer would have asked for a statistical analysis, and the authors have added it, but surely it was not this he/she was meaning. Therefore, it is highly recommended the removal of this section as its content is an absolute nonsense, which would hinder the acceptance of the manuscript for publication.

Still relevant are the recommendations concerning the conclusions.

Ln 480 should be something like “First of all, the increase in the level of BGWNP increased the AAMs working time as BGWNP incorporation has shown to delay the setting.”

Ln 482 is suggested to change for “Secondly, addition of 5% BGWNP was established to be most suitable to improve AAMs engineering properties. Still, for replacement level of 10% there was still engineering performance improvement in comparison with the reference mortar.”

Ln 486 The sentence shall be deleted as it becomes redundant.

If the authors believe that conclusions are too short, they may start them by recapping what replacement levels were investigated and what tests were carried out.

Minor criticisms and suggestions

Replacement of energy use by energy consumption throughout the text.

Replacement of tensile strength by splitting tensile strength throughout the text.

Ln 18 absorption instead of assimilation

Ln 43 and Ln 67 short setting timespan instead of rapid duration of setting

 Ln 72 Drying shrinkage instead of dry shrinkage

Ln 125 meaning of 16 ? 40 mm is not clear. Does it stand for 16 ball with 40 mm diameter?

Ln 144 However in the present work different values were considered

Ln 148 silicon dioxide:sodium oxide ratio

Ln 151 following ASTM C117.

Ln 201 The mean value of 3 samples from each AAM…

Ln 204 … as the mean of the results from 3 specimens of each AAM.

Ln 214 Water was eliminated from the AAMs surface to determine …

Ln 216 … via the total immersion method was computed through Eq. (1).

Ln 233 larger or smaller?

Ln 262 replacing GBFS instead of substitution of GBFS.

Ln 263 decreased instead of declined

Ln 272 initial and final instead of first and last

Ln 273 increased instead of were lengthened

Ln 274-5 ibidem

Ln 278-9 Please check, because it seems contradictory

Ln 281 fresh-state properties instead of qualities

Ln 288 how can one state that the increase is steady if there is no other measurement between 0 and 5%?

Ln 295-6 The sentence would be better if rephrased to something like “The results of [46,54] have shown improvements in mortar compressive strength for increasing levels of nano-silica up to 10% and decreasing compressive strength for concentrations higher than 10%.”

Ln 320 Lacking details for the procedures of this test in section 2

Ln 325 “… reduction not only in MoE, but also in FS and STS [21].”??!! Looking at figs. 7 and 8, for “AAMs with no more than 10% BGWNP”, an increase in FS and STS seems to exist.

Author Response

(The authors gave the same response as above.)
